# Adaptive Separation of Respiratory and Heartbeat Signals among Multiple People Based on Empirical Wavelet Transform Using UWB Radar

**DOI:** 10.3390/s20174913

**Published:** 2020-08-31

**Authors:** Mi He, Yongjian Nian, Luping Xu, Lihong Qiao, Wenwu Wang

**Affiliations:** 1School of Biomedical Engineering and Imaging Medicine, Army Medical University (Third Military Medical University), Chongqing 400038, China; yjnian@tmmu.edu.cn (Y.N.); xlp@tmmu.edu.cn (L.X.); 2College of Computer Science and Technology, Chongqing University of Posts and Telecommunications, Chongqing 400065, China; qiaolh@cqupt.edu.cn; 3Department of Electrical and Electronic Engineering, University of Surrey, Guildford GU2 7XH, UK; w.wang@surrey.ac.uk

**Keywords:** non-contact, radar, empirical wavelet transform (EWT), separation, vital sign

## Abstract

The non-contact monitoring of vital signs by radar has great prospects in clinical monitoring. However, the accuracy of separated respiratory and heartbeat signals has not satisfied the clinical limits of agreement. This paper presents a study for automated separation of respiratory and heartbeat signals based on empirical wavelet transform (EWT) for multiple people. The initial boundary of the EWT was set according to the limited prior information of vital signs. Using the initial boundary, empirical wavelets with a tight frame were constructed to adaptively separate the respiratory signal, the heartbeat signal and interference due to unconscious body movement. To verify the validity of the proposed method, the vital signs of three volunteers were simultaneously measured by a stepped-frequency continuous wave ultra-wideband (UWB) radar and contact physiological sensors. Compared with the vital signs from contact sensors, the proposed method can separate the respiratory and heartbeat signals among multiple people and obtain the precise rate that satisfies clinical monitoring requirements using a UWB radar. The detection errors of respiratory and heartbeat rates by the proposed method were within ±0.3 bpm and ±2 bpm, respectively, which are much smaller than those obtained by the bandpass filtering, empirical mode decomposition (EMD) and wavelet transform (WT) methods. The proposed method is unsupervised and does not require reference signals. Moreover, the proposed method can obtain accurate respiratory and heartbeat signal rates even when the persons unconsciously move their bodies.

## 1. Introduction

Heartbeat and respiration rates are two basic physiological indicators that reflect human health conditions and are normally monitored during clinical examination and treatment. Compared with traditional contact measurement, the non-contact clinical monitoring of vital signs is more convenient, safer and efficient, attracting considerable attention for applications in infectious disease wards, burn units, neonatal intensive care units, elderly home healthcare, and so on.

In recent years, numerous non-contact vital sign measurement systems, such as ultrasonic proximity sensors [1], infrared thermography [2], cameras and life radars [3,4], have been implemented for respiration or heartbeat monitoring. Given its intrinsic advantages, such as robustness to light, temperature and sound, life radar is more reliable for applications in such imperfect surroundings as hospital wards and the home environment [5]. Compared with continuous wave (CW) Doppler radar, ultra-wideband (UWB) radar, whose fractional bandwidth (i.e., bandwidth divided by the band centre frequency) is greater than 25% [6], can locate persons while recording vital sign signals, distinguish multiple persons in a room, and suppress clutter from different range bins [7,8]. According to different setups, UWB radar can be further categorized into impulse, frequency modulated continuous wave (FMCW), and stepped-frequency continuous wave (SFCW) UWB radar [2,9]. Impulse UWB radar has limited transmission power due to the narrow pulse and complex system of the analogue-to-digital converter (ADC) requirement. FMCW UWB radar may suffer from nonlinearity during frequency sweeping. Compared with impulse and FMCW UWB radar, SFCW UWB radar has a comparable range and velocity resolution but is a less complicated system and is therefore used for vital sign monitoring in this work.

Echoes signals of life radar sensors include mixed signals of respiration and heartbeat, which need to be further separated. Simple separation methods employ band-pass filters considering that respiration and heartbeat signals usually belong to different frequency bands. Due to interference from subtle movements and the variant frequency of the vital signs of different persons, fixed band-pass filtering will not always achieve accurate results [10]. As heartbeat signals are weaker than the respiratory signals by approximately one order of magnitude, the frequency component of a heartbeat signal may be merged by frequency spectrum leakage or higher harmonics of a respiratory signal. To separate such mixed signals, several adaptive filtering techniques have been developed, including wavelet independent component analysis (WICA) [3], ensemble empirical mode decomposition (EEMD) [11], complete ensemble empirical mode decomposition (CEEMD) [12] and blind source separation (BSS) [13]. In WICA, wavelet decomposition and independent component analysis are combined to decompose the mixed vital signs and to estimate the heartbeat signal. The largest variability in root mean-square errors for the rate of the separated heartbeat signal was reported to be 3.328 bpm [3]. A two-layer pseudo bi-dimensional EEMD was used to extract the heartbeat signal in [11], which, however, did not include a comparison with reference vital signals. In addition, EEMD does not always provide decomposition of signals into components of desired frequency bands. A non-negative matrix factorization (NMF) algorithm-based BSS technique was proposed in [13] to reduce the noise from the received radar signal, where sparse spectrum reconstruction and the zero-attracting sign least-mean-square were incorporated to estimate the heartbeat spectrum with an average absolute error of 3.18 bpm for five volunteers. These methods are, however, complicated and computationally expensive.

To address these limitations, in this paper, we propose a novel vital sign separation method based on empirical wavelet transform (EWT) using SFCW-UWB radar. EWT is an adaptive signal analysis approach that is suitable for dealing with nonlinearity in the model and non-stationary signals with elegant mathematical theory [14]. In recent years, EWT has been successfully applied to medical disease diagnosis, image processing, machine fault diagnosis, and seismic data analysis [14]. In our work, prior information of the relatively stable frequency ranges of respiration and heartbeat signals is adopted to derive the EWT boundaries, which are then used to separate respiration and heartbeat signals without the need to employ other referenced signals. The proposed method has a compact architecture that is adapted to separate the vital signs of multiple human targets from the UWB radar. The respiratory and heartbeat signals obtained by radar are compared with the vital signs from contact sensors. To our knowledge, this is the first work to use EWT to extract respiratory and heartbeat information from radar signals and to acquire vital signs that satisfy the clinical requirements for multiple targets in a non-scanning mode.

The remainder of this paper is organized as follows. Section 2 presents a block diagram of the signal processing procedure, the signal model of the SFCW-UWB radar and the theory of the proposed algorithm based on EWT. In Section 3, the experimental scenario and data processing parameters are described. Section 4 shows the experimental results from volunteers in a room and presents an analysis of the results. The conclusion is summarized in Section 5.

## 2. Signal Model and Proposed Algorithm

### 2.1. Block Diagram of Signal Processing

Figure 1 shows the whole signal processing flowchart of the proposed algorithm for SFCW-UWB radar vital sign monitoring.

#### 2.1.1. Acquire the Beat Signal

The transmitted signal (i.e., a series of frames) is multiplied by the complex conjugate of the received signal to obtain the beat signal, and this step is termed de-ramping processing [15]. The beat signal is converted to a baseband signal with a narrow bandwidth while still containing all the information about the targets.

#### 2.1.2. Obtain the Range-Time Matrix

Next, the inverse Fourier transform is applied to the beat signal at each frame to realize range compression. In practical implementation, the inverse fast Fourier transform (IFFT) is applied on the *v*-th (*v* = 1, 2, …, *V*) frame. After IFFT, a range-time complex matrix is obtained as *RS(h, v)*, where the index *v* (=1, 2, …, *V*) is defined as the slow time direction, and the index *h* (=1, 2, …, *H*) is defined as the fast time direction [2].

#### 2.1.3. Acquire the Range Information from the Targets

For the suppression of the stationary clutter due to floor and furniture, the moving target indication (MTI) technique [16] is applied to the range-time matrix *RS(h, v)* along the time dimension to obtain RS_MTI_(h, v). MTI is a common operation mode used in radar to discriminate a target from stationary clutter [16]. The absolute value of each row of the range-time matrix is called the high range resolution profile (HRRP). For the suppression of sidelobes from the HRRP, a Hanning window is implemented before applying IFFT. For each quasi-static human target, the range bin location in each scheme can be considered fixed. Thus, the range bin locations (or range information) of multi-targets can be detected from the HRRP using MTI. Constant false alarm rate (CFAR) detection is an adaptive algorithm that is widely used in radar systems to detect targets in a background of noise, clutter and interference. Here, a computationally efficient cell-averaging CFAR detector is applied on each HRRP after MTI. More details of MTI and CFAR can be found in [17].

#### 2.1.4. Obtain Mixed Vital Sign Signals of Targets

The phase of the *m*-th human target is demodulated and unwrapped from the *h_m_*-th column vector of *RS(h, v)* and denoted *RS* (*h_m_, v*), *m* = 1, 2, …, *M* and *v* = 1, 2, …, *V*. The direct current (dc) component is then subtracted from the phase signal of each human target. The remaining phase signal denoted as $\Delta R_{m}\left(\tau \right)$ (seen in Equation (5)) represents the mixed vital sign signal of the *m*-th human target, where τ is defined as the continuous slow time.

#### 2.1.5. Acquire the Fundamental Respiration Frequency of Targets

$\Delta R_{m}\left(\tau \right)$ is passed through a high-pass filter with a cutoff frequency of 0.1 Hz, resulting in $\Delta R_{m}^{\prime }\left(\tau \right)$. The Fourier transform is applied on $\Delta R_{m}^{\prime }\left(\tau \right)$ to obtain the frequency spectrum as $F_{\Delta R_{m}}\left(f\right)$. The frequency at the maximum of $F_{\Delta R_{m}}\left(f\right)$ is obtained and denoted *f_p_*. If *f_p_* is within the respiration frequency range of healthy adults, i.e., [0.15 0.40] Hz [18], and there is a peak (i.e., the second harmonic of respiration) in the frequency range [(2 − 0.2) *f_p_* (2 + 0.2) *f_p_*] Hz, the fundamental respiration frequency *f_R_*_0_ can be estimated as equal to *f_p_*; otherwise, the fundamental respiration frequency must be further determined.

#### 2.1.6. Construct Initial Boundary and Empirical Wavelet Functions

In the proposed method, the initial boundary of the EWT is very important to determine the scale segment in the frequency domain. Here, the initial boundary of the EWT is constructed as [*f_R_*_0_, 2 *f_R_*_0_, 3 *f_R_*_0_, 4 *f_R_*_0_] × *V/f_s_*, where *V* is the slow time sampling number, and *f_s_* is the slow time sampling frequency. If the fundamental respiration frequency is not found, an empirical value, e.g., 0.3 Hz, is set for *f_R_*_0_. The final boundary of EWT is calculated adaptively according to the frequency characteristic of the mixed signal. The scale function and the wavelet functions of the EWT in the frequency domain are constructed by using a 1-D Meyer wavelet according to the final boundary and the tight frame condition [19]. After taking the inverse Fourier transform, we obtain the scale function and wavelet functions of the EWT in the time domain.

#### 2.1.7. EWT Decomposition and Vital Sign Signal Separation

The mixed vital sign $\Delta R_{m}\left(\tau \right)$ is decomposed by inner products with empirical wavelets. After obtaining the first layer of EWT decomposition, the frequency corresponding to the maximum of each component is calculated. If the fundamental respiration frequency *f_R_*_0_ is already determined, the EWT component including the frequency content *f_R_*_0_ is considered the separated respiratory signal; otherwise, the EWT component whose maximum frequency is located in the respiration frequency range of healthy adults in the remaining components, i.e., [0.15–0.40] Hz, is selected as the separated respiratory signal [18]. Analogously, among the remaining EMT components, the one whose maximum frequency is located in the range of [0.80–1.68] Hz, i.e., the heartbeat frequency range of healthy adults, is chosen as the separated heartbeat signal [18]. Usually, the first or second EWT component contains the respiratory information, and the third or fourth EWT component may include harmonics of the respiratory signal and the heartbeat signal in the proposed method. Another EWT layer can be executed to further refine the frequency content of the separated respiration and heartbeat signals. Finally, the range information *h_m_* and the extracted respiration and heartbeat signals for each human target are obtained.

### 2.2. Signal Model of the SFCW-UWB Radar

Assume *M* human targets are simultaneously illuminated by the SFCW-UWB radar. According to [2,20], the inverse Fourier transform of the down-converted signal in each frame of the SFCW radar is given by:(1)$$\begin{array}{ll} S_{b}\left(f,\tau \right)= & \sum_{m=1}^{M}\frac{A_{m}}{T}\cdot \exp\left(\frac{j4\pi f_{0}R_{m}\left(\tau \right)}{c}\right)\\ & \cdot \exp\left(\frac{j2\pi (N-1)}{N}\left(f-\frac{2N\Delta fR_{m}\left(\tau \right)}{c}\right)\right)\\ & \cdot \sin c\left(T\left(f-\frac{2N\Delta fR_{m}\left(\tau \right)}{c}\right)\right) \end{array}$$
where *f* is the frequency along the fast time direction, *A_m_* is the amplitude of the *m*-th human target reflection, *T* is the repetition time of one pulse, each frame of the transmitted signal includes *N* pulses with the initial carrier frequency *f*_0_ and the linearly increased frequency step Δ*f*, the sampling interval of the continuous slow time τ is *N* × *T*, c is the speed of light, $R_{m}\left(\tau \right)$ is the range of the *m*-th human target from the radar as a function of the slow time, and the function $\sin c\left(x\right)=\frac{\sin\left(\pi x\right)}{\pi x}$. The function $R_{m}\left(\tau \right)$ can be written as:(2)$$R_{m}\left(\tau \right)=R_{m}^{0}+\Delta R_{m}\left(\tau \right)$$
where $R_{m}^{0}$ is the static range, and $\Delta R_{m}\left(\tau \right)$ is the time-varying range of chest wall vibrations due to the respiration and heartbeat between the radar and the *m*-th human target.

Suppose the vibration of human chest walls along the line-of-sight (LOS) is no greater than the range resolution, the scatters remain in their corresponding range during the coherent processing intervals. According to Equation (1), the range information in the frequency domain of the *m*-th target is:(3)$$f_{m}=\frac{2N\Delta fR_{m}\left(\tau \right)}{c}$$

At frequency *f_m_*, the range spectrum (i.e., the absolute of Equation (1)) achieves its peak value and Equation (1) equals:(4)$$S_{b}\left(f_{m},\tau \right)=\frac{A_{m}}{T}\cdot \exp\left(j\frac{4\pi f_{0}}{c}R_{m}\left(\tau \right)\right)$$

Thus, the mixed vital sign signal $\Delta R_{m}\left(\tau \right)$ of the *m*-th target can be recovered by unwrapping the demodulated phase signal along a slow time:(5)$$\Delta R_{m}\left(\tau \right)=\frac{c}{4\pi f_{0}}\cdot \mathrm{unwrap}\left[s_{b}\right]-\mathrm{mean}(\frac{c}{4\pi f_{0}}\cdot \mathrm{unwrap}\left[s_{b}\right])$$

Demodulated phase signals of *M* human targets can be considered independent. The theoretical values of the range resolution and the maximum ambiguity range of the SFCW radar are $\Delta r=\frac{c}{2B}$ and $\Delta r_{MA}=\frac{c}{2\Delta f}$, respectively, where *B* (=*N*Δ*f*) is the bandwidth of the SFCW radar.

### 2.3. Theory of the Proposed Algorithm Based on EWT

For EWT, how to set the boundaries on the Fourier spectrum is very important because this setting provides adaptability with respect to the analyzed signal. According to prior knowledge [18], the frequencies of the heartbeat signals are usually similar to or higher than the third harmonic of the respiratory signals.

Let *ω_k_* denote the *k*-th centre of the *K* contiguous frequency segments corresponding to the EWT components, where *ω*_0_ = 0 and *ω_k_* = π. In the frequency domain, the empirical scaling function ${\hat{\phi }}_{k}\left(\omega \right)$ and the empirical wavelets ${\hat{\psi }}_{k}\left(\omega \right)$ on each frequency segment can be constructed according to [19]. The empirical wavelet basis as a set $\left\{\phi_{1}\left(t\right),{\left\{\psi_{k}\left(t\right)\right\}}_{k=1}^{K}\right\}$ is a tight frame of Euclidean norm space *L*_2_(*R*) when the transition ratio parameter γ is chosen as $\gamma <\min_{k}\left(\frac{\omega_{k+1}-\omega_{k}}{\omega_{k+1}+\omega_{k}}\right)$ [19].

The detailed coefficients of $\Delta R_{m}\left(\tau \right)$ are calculated by the inner products with the empirical wavelets as:(6)$$D_{m}\left(k,\tau \right)=\langle \Delta R_{m}\left(\tau \right),\psi_{k}\left(\tau \right)\rangle =\int \Delta R_{m}\left(\tau \right)\psi_{k}^{*}\left(t-\tau \right)dt$$
where * represents conjugate.

The approximation coefficient of $\Delta R_{m}\left(\tau \right)$ is given by the inner products with the empirical scaling function:(7)$$A_{m}\left(0,\tau \right)=\langle \Delta R_{m}\left(\tau \right),\phi_{1}\left(\tau \right)\rangle =\int \Delta R_{m}\left(\tau \right)\phi_{1}^{*}\left(t-\tau \right)dt$$

The reconstruction of $\Delta R_{m}\left(\tau \right)$ is expressed by:(8)$$\begin{array}{ll} \Delta R_{m}\left(\tau \right) & =\Delta R_{m}^{0}\left(\tau \right)+\sum_{k=1}^{K}\Delta R_{m}^{k}\left(\tau \right)\\ & =A_{m}\left(0,\tau \right)\phi_{1}\left(\tau \right)+\sum_{k=1}^{K}D_{m}\left(k,\tau \right)\psi_{k}\left(\tau \right) \end{array}$$
where $\Delta R_{m}^{k}\left(\tau \right)$ (*k* = 0,1,…,*K*) are the (*k* + 1)-th EWT component with a centre frequency *ω_k_*.

Suppose *ω*_1_ is the centre frequency of the first frequency band containing the respiration fundamental frequency, the second harmonic frequency of the respiratory signal may exist in the second frequency band with the centre frequency 2*ω*_1_. Similarly, the third harmonic of the respiratory signal belongs to the third frequency band with the centre frequency 3*ω*_1_. Moreover, the frequency of the heartbeat signal usually belongs to the third frequency band as well. The above prior knowledge can be used to initialize the frequency boundaries for EWT. 

The detail of the adaptive calculation of final boundaries of EWT is as follows. First, some neighbourhood on the frequency domain is computed from the initial boundaries. Next, the global minima value in each neighbourhood is detected. Last, the final boundaries are computed using the input mixed signal and the location of the minima.

## 3. Experiment

### 3.1. UWB Radar Platform

The experimental setup for non-contact multi-target vital sign monitoring using the SFCW radar is depicted in Figure 2, including a vector network analyzer (VNA, Rohde & Schwarz, Munich, Germany, ZNB20), a pair of standard X-band UWB horn antennas (Xi’an Hengda, Xi’an, China, HD-100SGAH15N), a computer (Lenovo, Beijing, China, Inter Core i5-4460, CPU 3.2 GHz), finger-clip pulse oxygen sensors (Goldway, Shenzhen, China, UT4000) for the photoplethysmography (PPG) signals, breathing belt sensors (Huake, Hefei, China, HKH-11B) for respiratory signals and a multi-channel data acquisition board (DAB, National Instruments, Austin, TX, USA, USB-6341). The VNA functions as a signal generator at the 0 dBm power level, a signal receiver and a partial signal analyzer that provides DC offset, down conversion and unwrapping. The VNA is connected by two cables to the transmitting and receiving X-band antennas with a beam width of 26° and a gain of 16.1 dB at the central frequency of 10.35 GHz. A pair of antennas with a beam pointing elevation angle of 120° are fixed on the antenna server system (ASS), where the elevation angle was defined as 90° when the antennas were facing the floor and as 0° when the antennas were facing the windows and horizontal to the ceiling.

As shown in Figure 2, the VNA, the DAB and the ASS are all controlled by a computer. The sampled radar signals by VNA are sent and saved to the computer for further signal processing. To ensure the synchronization of the VNA and the physiological contact sensors, the trigger pulse signal of each frame from the VNA is sent to the DAB. The physiological signals of multiple persons and the trigger signal were simultaneously sampled by the DAB and then sent to the computer. Before running the experiment, the VNA with cables, the VNA with antennas, the ASS, the DAB and the contact sensors were all carefully calibrated.

### 3.2. Participants

Three healthy males (aged from 22 to 38 years) were recruited for this study. All volunteers signed the informed consent and were given instructions about the experiment and attached contact sensors before measurements. They were asked to be relaxed and maintain a static posture as much as possible during the measurements. Figure 3 presents the actual experimental scenario. Two volunteers were lying on an air mat, and one volunteer was lying on a wooden bed. The distance between the floor and the antennas was approximately 2.3 m. The ranges between the radar antennas and the three volunteers are different so that the multiple targets could be distinguished from the received radar signals.

### 3.3. Data Acquisition and Analysis

In the frequency sweeping model of the VNA, the initial carrier frequency was set to 8.35 GHz, and the end carrier frequency was set to 12.35 GHz with an increasing frequency step of 20 MHz. The sweeping frequency range was consistent with the bandwidth of the X-band UWB antenna. The frequency sweeping period was 0.05 s, i.e., the slow time sampling frequency was 20 Hz, which is sufficient for the slowly varying respiratory and heartbeat signals. The fast sampling frequency was 2400 Hz. To capture the trigger signal of the VNA and to maintain synchronization, the sampling frequency of the DAB was set to 2 kHz. Moreover, the theoretical values of the range resolution and the maximum unambiguity range of the SFCW radar were 3.75 cm and 7.5 m, respectively. Moreover, the actual experimental range resolution was 3.75 cm by using the CFAR detection. For consistency, the physiological signals were down sampled to 20 Hz. All time delays of the physiological signals due to cables were carefully compensated for according to the pretest. In the experiment, each measurement lasted 2 min and was repeated 10 times.

For comparison, bandpass filtering, EMD and wavelet transform (WT) were applied to separate the mixed vital sign signals for each volunteer. The bandpass filter was designed with a start frequency of 0.1 Hz and stop frequency 0.7 Hz for the respiratory signals, and with a start frequency 0.7 Hz and a stop frequency of 2 Hz for the heartbeat signals. As EMD is prone to noise, the mixed vital sign signals are first passed through a band-pass filter before EMD. The intrinsic mode function (IMF) with the highest frequency component, which is usually presented as IMF_1_, contains much interference due to the presence of the respiration harmonics. To extract the heartbeat signal, IMF_1_ was passed through a high-pass filter with a cutoff frequency of 0.7 Hz. IMF_2_ can reflect the respiratory signal with only the fundamental frequency. For the WT method, 6-level wavelet decomposition was applied to the mixed signals. The first or second layer decomposition component was extracted as the respiratory signal; the third layer detail coefficients passed through a high-pass filter with a cutoff frequency of 0.7 Hz were extracted as heartbeat signals. Considering the frequency characteristics of the vital signs and the computation complexity, the maximum number of components or frequency bands was set to 5 in the proposed method.

To assess the agreement between two quantitative methods or systems of measurement, the Bland–Altman analysis was applied to the respiration and heartbeat frequency by different methods. The Bland–Altman analysis is an effective method for determining the limits of agreement between measurements using two different methods [21]. The differences in physiological signal measurements were visualized using a Bland–Altman plot, which depicts the bias between the physiological signals recorded from the contact sensors and the physiological signals extracted by radar using a separation method with respect to the average of these two signals. The Bland–Altman plot is a scatter diagram, where the *X* axis corresponds to the average of the two measures, and the *Y* axis represents their difference [22]. The 95% confidence agreement interval is also shown on the scatter plot. The statistical agreement limits are calculated by using the mean (MD) and the standard deviation (SD) of the differences between two measurements [22]. Within the agreement interval, 95% of the differences will be between MD − 1.96SD and MD + 1.96SD if the differences are normally distributed.

## 4. Results and Discussion

### 4.1. Range-Time Spectrogram of Volunteers

The original range-time spectrogram and the range-time spectrogram after applying the MTI are presented in Figure 4. Both spectrograms were normalized to decibels (dB). There was much clutter in the original range-time spectrogram due to the floor and furniture, which interfered with the physiological signals from the volunteers (as seen in Figure 4a). Fortunately, almost all the static clutter was suppressed after applying the MTI while the vital sign information was retained, as shown in Figure 4b. The range bins of the three volunteers can be verified according to the range-time spectrogram after applying the MTI.

To further show the determination of range bins for multiple targets, the HRRP at 25.35 s before and after applying MTI was taken as an example, as shown in Figure 5. Similarly, the original HRRP was corrupted by clutter, which resulted in the incorrect location of the targets. The clutter disappeared from the HRRP after applying the MTI while the targets were still prominent, although the relative amplitude was decreased. Next, the range bins of the multiple targets were detected by the CFAR detector on the HRRP after MTI with the false alarm 0.001 and the guard cell number 6. The range bins of volunteer 1, volunteer 2 and volunteer 3 were 42, 59 and 66, respectively. Accordingly, the ranges of volunteer 1, volunteer 2 and volunteer 3 from radar were 1.575 m, 2.213 m and 2.475 m, respectively.

### 4.2. EWT Composition

Figure 6 presents the initial (thin black dashed line) and final (bold red dashed line) boundaries of the EWT (i.e., the Fourier supports) on the mixed frequency spectrum for volunteer 2 in the first measurement. The result in Figure 6 shows that the initial boundaries indeed guided the final boundaries. Five contiguous frequency segments were divided by the final boundaries for further EWT decomposition.

The EWT components with respect to the contiguous frequency segments are shown in Figure 7. The second (Figure 7b), third (Figure 7c) and fourth (Figure 7d) EWT components were the respiratory signal with the fundamental frequency, the second respiration harmonic and the heartbeat signal, respectively. The abscissa and ordinate values of the maximum peak for the second, third and fourth EWT components were noted on their frequency spectra, as shown in Figure 7g–i, respectively. The first component of the EWT (Figure 7a) came from low frequency interference. Figure 7f shows the corresponding frequency spectrum. The last component of the EWT (Figure 7e) with its frequency spectrum, as shown in Figure 7j, came from high frequency interference.

Moreover, the extracted respiratory and heartbeat signals by the proposed method were compared with the physiological signals acquired by the breathing belt and the finger-clip pulse oxygen sensors in Figure 8. For ease of observation, the 20-s respiratory signal and the 10-s heartbeat signal are presented as examples in Figure 8a,c, respectively. The unit of the physiological signals in Figure 8a,c is cm on the left *y* axis; the unit of the contact sensor signals in Figure 8a,c is volt on the right *y* axis. Additionally, the frequency spectra of the extracted signals from radar and the physiological signals from the contact sensors were compared, as shown in Figure 8b,d. Note that the frequency spectra here contained no dc signals. 

The results show that the respiratory and heartbeat signals extracted by the proposed method matched well with the physiological signals recorded from the contact sensors, especially in the frequency domain. The minor differences between the extracted heartbeat signal and the PPG signal were probably due to different body parts and detection media. The extracted heartbeat signal was derived from the chest vibration caused by cardiac systole and diastole based on microwaves, while the PPG signal was derived from the peripheral artery of the fingertip based on light waves. Pulse oximetry is a noninvasive technique for monitoring PPG signal that can reflect the cardiac motor state. Thus, the heartbeat rate can be approximately calculated from the PPG signal [23].

### 4.3. Comparison of Different Methods

If persons unconsciously moved their bodies slightly during the measurements, the interference would present as a strong low-frequency component on the frequency spectrum of the mixed vital sign from radar and similarly on the frequency spectrum of the PPG signal from the pulse oxygen sensor, as shown in the black box on the left side of Figure 9a. The respiratory frequency peak of the volunteer was not apparent in Figure 9a compared with the interference frequency, which may lead to an incorrect judgement of the respiratory frequency by traditional methods. Similarly, the pulse frequency peak was much lower than the interference frequency peak, as seen in the frequency spectrum of PPG signal in Figure 9a. Thus, the PPG signal was passed through a 0.1–2.5 Hz bandpass filter to obtain the filtered PPG signal for further analysis. The frequency spectrum of the filtered PPG signal (blue dash) was also shown in Figure 9a. The proposed method was compared with the bandpass filtering, EMD and WT methods. The relative parameters were provided in Section 3.2. Figure 9b,c showed that the proposed method can extract a more accurate respiration fundamental frequency than the bandpass filtering, EMD and WT methods. Similarly, the proposed method can distinguish the heartbeat frequency more precisely than the bandpass filtering, EMD and WT methods, as shown in Figure 9d,e. For convenience of comparison, the frequency spectra in Figure 9 contained no dc signals.

Ten continuous measurements were taken for the three volunteers. The respiratory and heartbeat rates were calculated using the extracted signals by different separation methods. The bias of the measured rates between the physiological signals recorded from the contact sensors and the extracted vital signal from radar using each separation method were presented in the Bland–Altman analysis in Figure 10, which included all the measurements. The horizontal axis corresponded to the average respiratory rate (RR) or heartbeat rate (HR) as (RR_CS_ + RR_Radar_)/2 or (HR_CS_ + HR_Radar_)/2 between the physiology signals measured from the contact sensors (CS) and radar using one separation method. Similarly, the vertical axis corresponded to the difference in RR or HR as (RR_CS_ − RR_Radar_) or (HR_CS_ − HR_Radar_). For the RR, the proposed method achieved the smallest MD of −0.021 bpm and the smallest SD of 0.099 bpm among the four methods. The largest bias in the RR was observed for use of the EMD method, with an MD of 1.636 bpm and an SD of 4.740 bpm. Moreover, the proposed method achieved the smallest MD of −0.084 bpm and SD of 0.6613 bpm for the HR estimation. The largest MD for the HR was obtained by the EMD method (1.992 bpm), and the largest SD was obtained by the band-pass filtering method (4.902 bpm). Table 1 listed the error estimation of RR and HR of three volunteers for different methods. The results were in accordance with those in Figure 10. All error statistical parameters of the proposed method were smaller than other methods. For the proposed method, the largest error of calculated RR was within ±0.3 bpm, which met the limits of agreement of within ±2 bpm based on clinical judgement [9]; the largest error in the calculated HR was within ±2 bpm, which was acceptable according to the American National Standard ANSI/AAMI EC13: 2002 (within ±5 bpm) [24].

The wavelet decomposition component derived from the proper level of detail coefficients will produce the same frequency peak due to the fixed wavelet basis [25]. Thus, the frequencies of the respiratory or heartbeat signals calculated by the WT method may not be correct. In [3], all wavelet decomposition components of the mixed phase signal of radar were input into an ICA analysis for further separation. For 12 measurements taken from one volunteer, the maximum root mean-square error of the HR was 3.19 bpm [3]. However, this method was not suitable for the mixed phase signal with interference. EMD and EEMD separate signals according to IMFs, which are not always able to acquire components in specific frequency bands [11]. In [13], NMF of the spectrograms of the vital signals solved an optimization problem with two parameters, which may only yield a local optimal solution. For measurements from five subjects without repetition, the smallest average HR error is 3.18 bpm [13]. A previous study reported that employing a phase array antenna in an impulse UWB radar to scan two volunteers at different elevation angles achieves an error of 2.85% for HR detection [26]. The proposed method yields a higher accuracy in measuring RR and HR than the methods reported in the above publications and satisfies current clinical requirements. A limitation in this work is the use of data from a limited number of subjects, which will be further expanded in future clinical research.

## 5. Conclusions

The proposed method based on EWT can separate the mixed phase from radar into the respiratory signal and the heartbeat signal even when the persons unconsciously move their bodies by comparison with physiological signals recorded from contact sensors. The estimated respiratory rate and heartbeat rate errors using the proposed method were within ±0.3 bpm and ±2 bpm, respectively, which are much lower than the errors associated with existing methods, including band-pass filtering, EMD and WT methods. Using the proposed method with SFCW-UWB radar, the respiratory and heartbeat signals of multiple people can be simultaneously extracted and separated with high accuracy. Moreover, the proposed separation method can also be applied to other types of UWB radars for multi-human target detection, and to CW radar for human target monitoring.

## Figures and Tables

**Figure 1 sensors-20-04913-f001:**
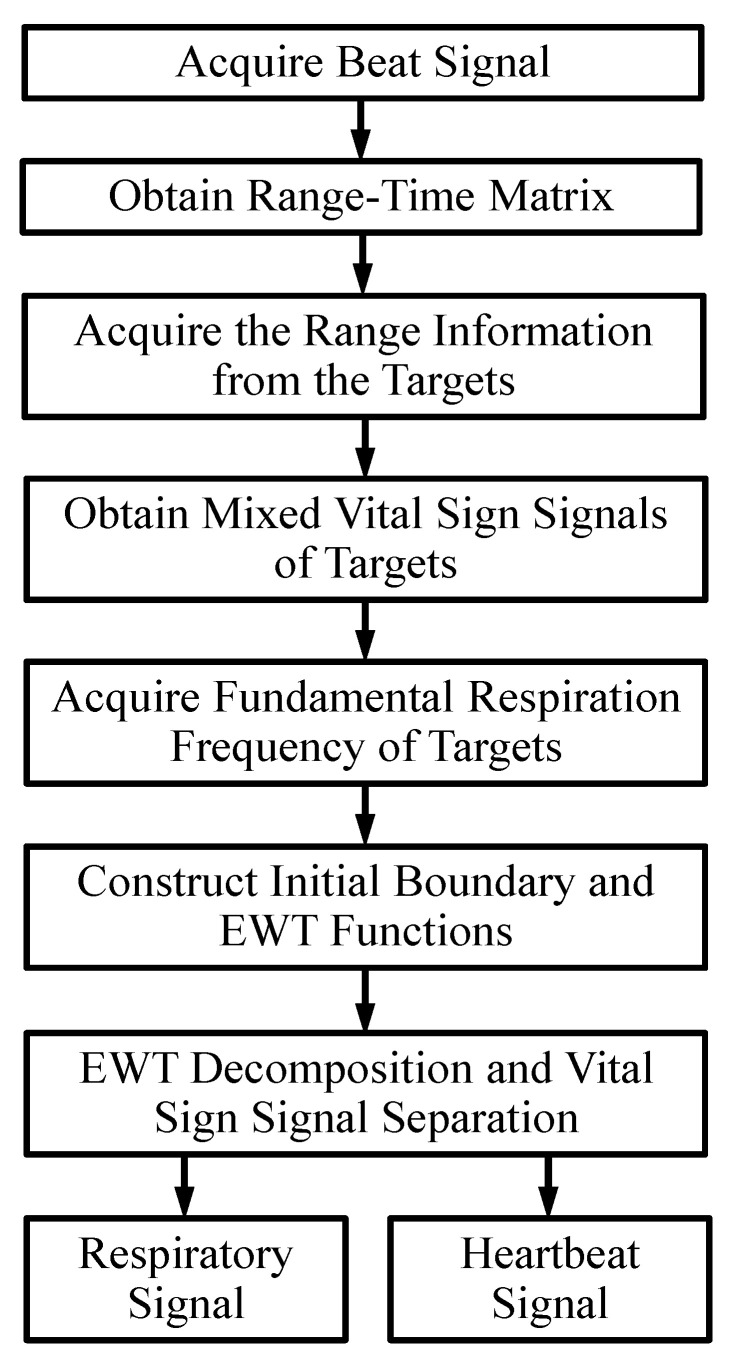
Signal processing flowchart of the proposed algorithm for multi-human vital sign separation from the SFCW-UWB radar measurements.

**Figure 2 sensors-20-04913-f002:**
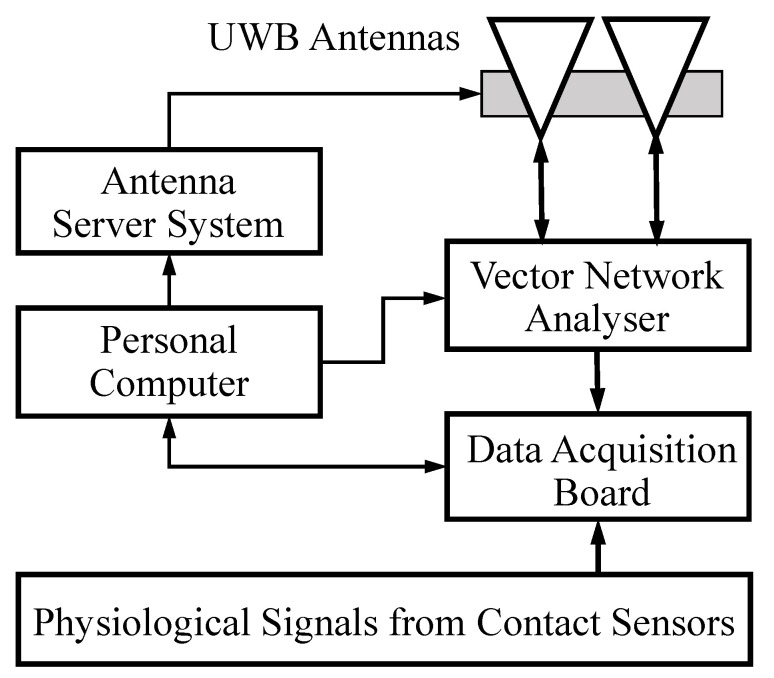
Experimental setup for non-contact multi-target vital sign monitoring; the physiological contact sensors include pulse oxygen sensors and breathing belt sensors.

**Figure 3 sensors-20-04913-f003:**
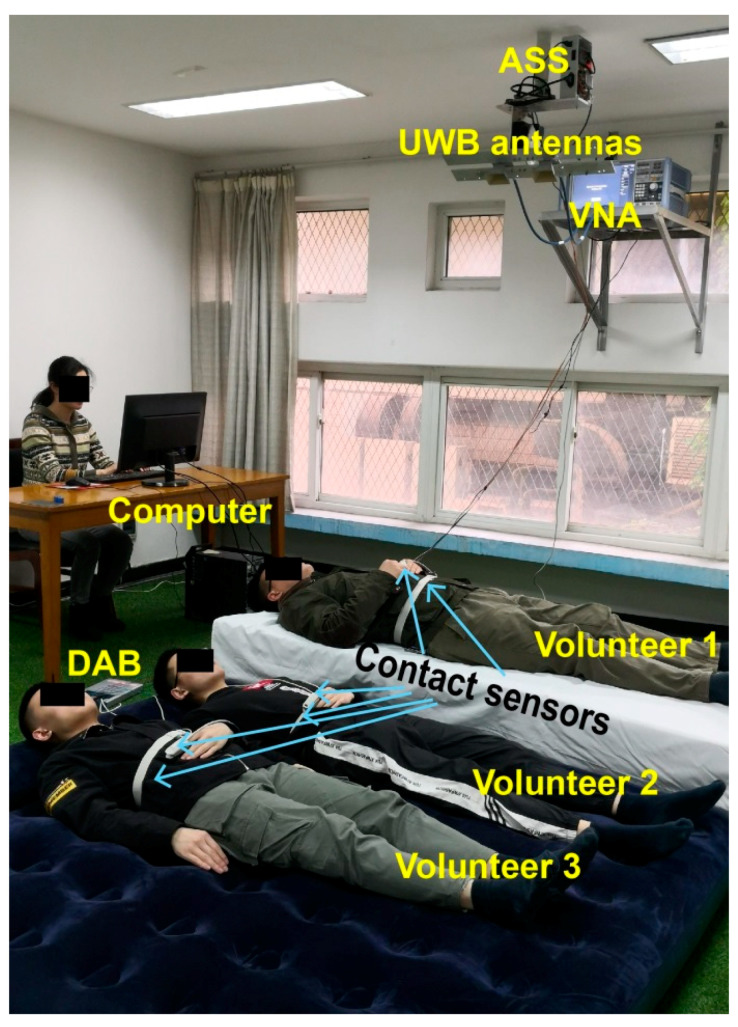
Actual experimental scenario with three volunteers in a room. ASS: antenna server system; VNA: vector network analyzer; DAB: data acquisition board.

**Figure 4 sensors-20-04913-f004:**
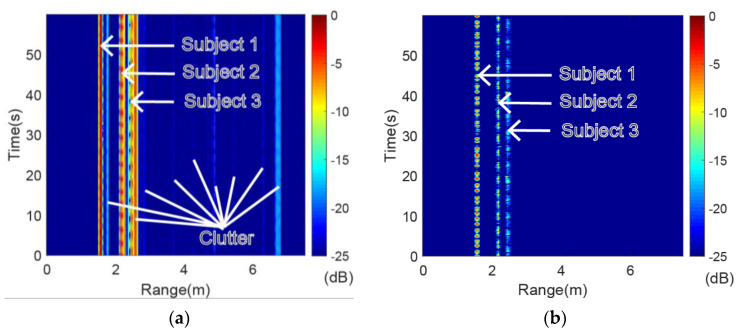
Normalized range-time spectrogram of three volunteers. (**a**) range-time spectrogram before moving target indication (MTI); (**b**) range-time spectrogram after MTI.

**Figure 5 sensors-20-04913-f005:**
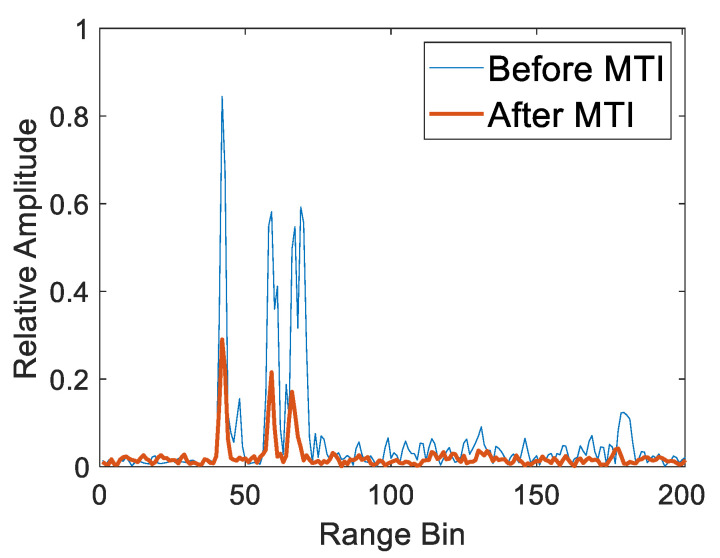
High range resolution profile (HRRP) of the three targets at 25.35 s before and after MTI.

**Figure 6 sensors-20-04913-f006:**
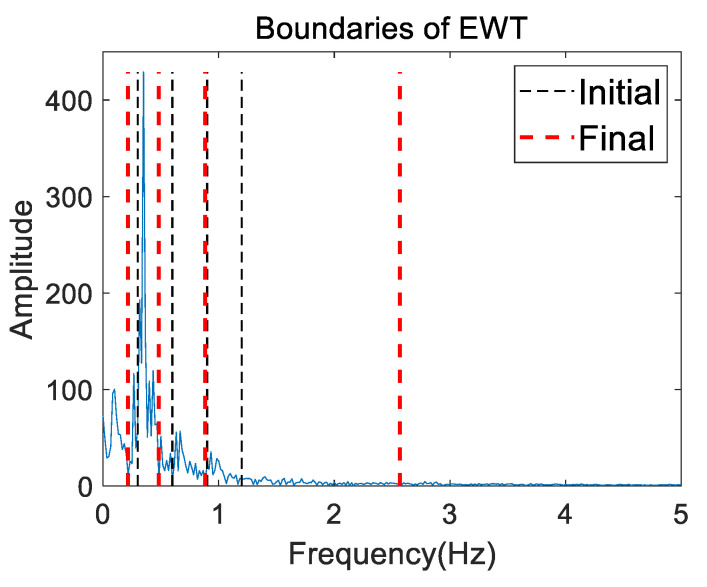
Initial and final boundaries of the empirical wavelet transform (EWT) on the mixed frequency spectrum for volunteer 2 in the first measurement.

**Figure 7 sensors-20-04913-f007:**
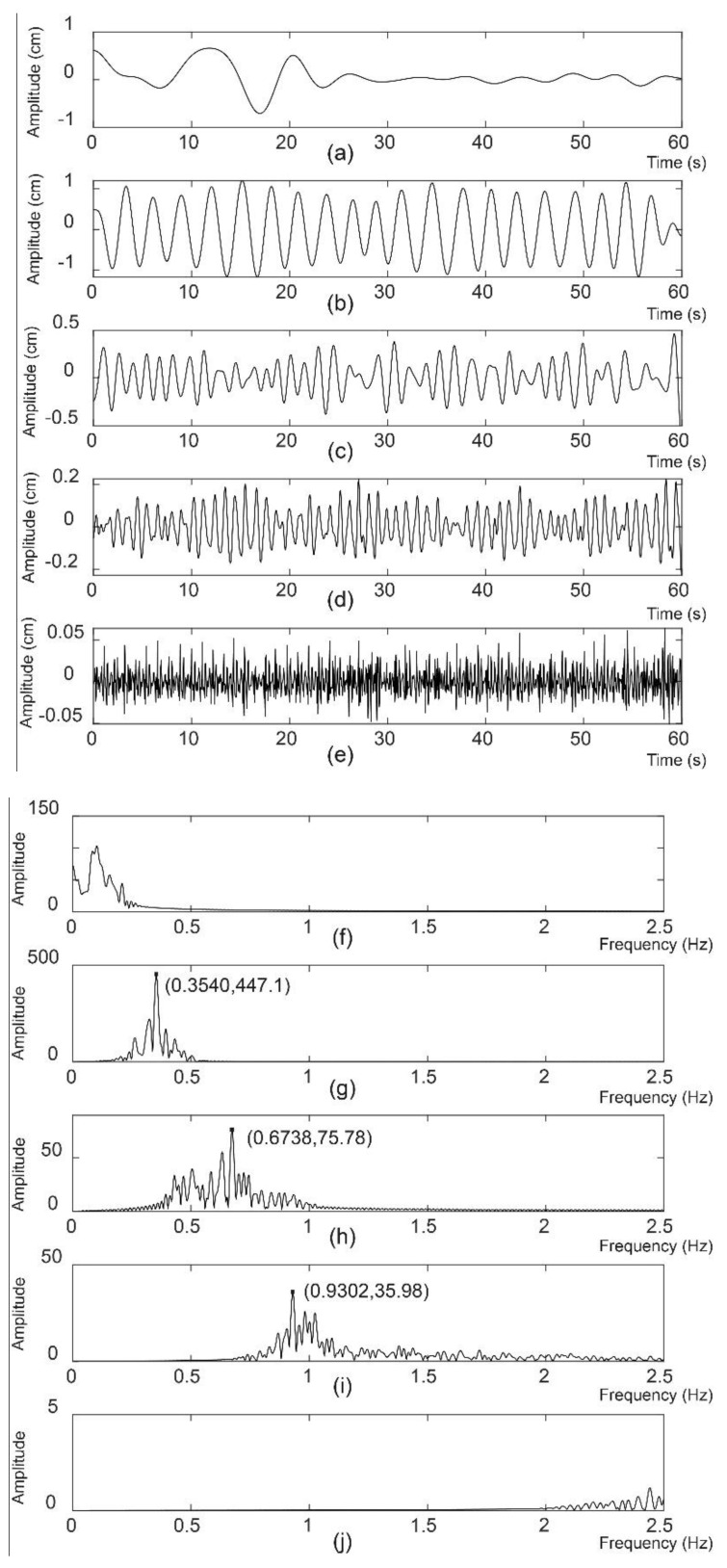
Five EWT components and their frequency spectra for volunteer 2 in the first measurement. (**a**–**e**) are the first, second, third, fourth and fifth EWT components in the time domain. The corresponding frequency spectra are shown in the right column (**f**–**j**).

**Figure 8 sensors-20-04913-f008:**
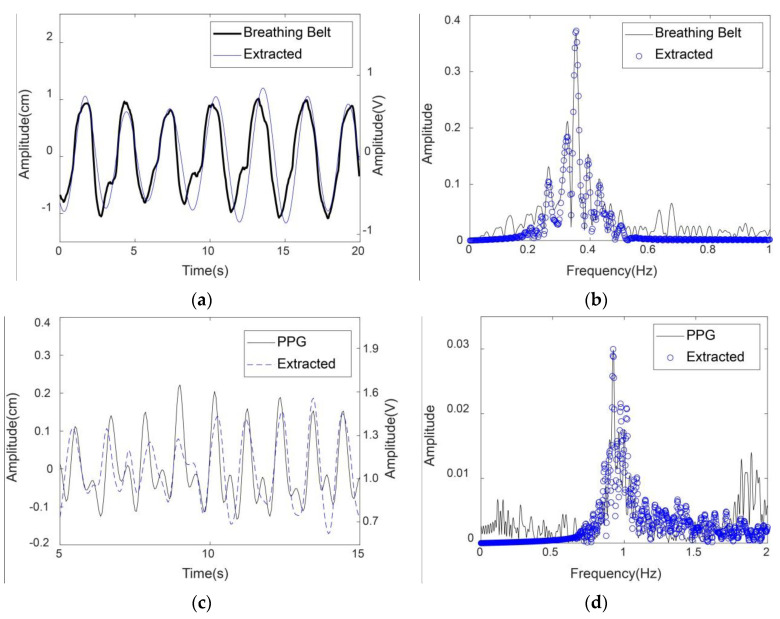
Comparison between the vital signs extracted from radar and the physiological signals recorded from contact sensors of volunteer 2 in the first measurement. (**a**,**b**) are the respiratory signals in the time domain and the frequency domain, respectively. (**c**,**d**) are the heartbeat signals in the time domain and the frequency domain, respectively. The frequency spectra contained no dc signals.

**Figure 9 sensors-20-04913-f009:**
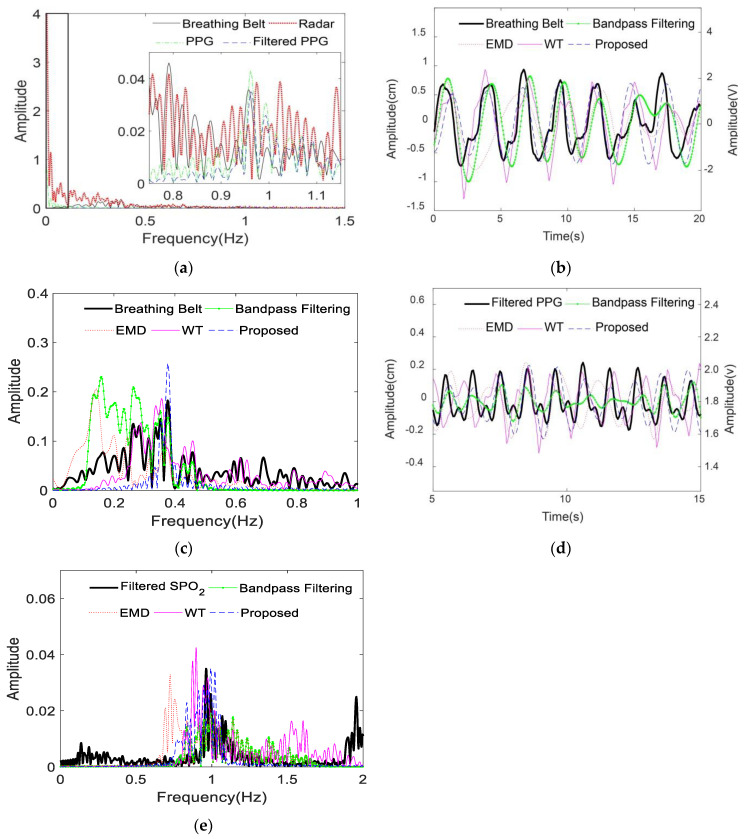
Comparison of the extracted vital signals from radar among the traditional and the proposed methods based on the physiological signals recorded from contact sensors for volunteer 2 in the sixth measurement. (**a**) shows the frequency spectra of the respiratory signal from the breathing belt, the mixed vital sign from radar, and the photoplethysmography (PPG) signal and filtered PPG signals from the pulse oxygen sensors; (**b**,**c**) show the respiratory signals from the breathing belt analyzed by the different separation methods in time and frequency domains, respectively; (**d**,**e**) show the filtered PPG signal and heartbeat signals analyzed by the different separation methods in the time and frequency domains, respectively. EMD: empirical mode decomposition method; WT: wavelet transform method. The frequency spectra contained no dc signals.

**Figure 10 sensors-20-04913-f010:**
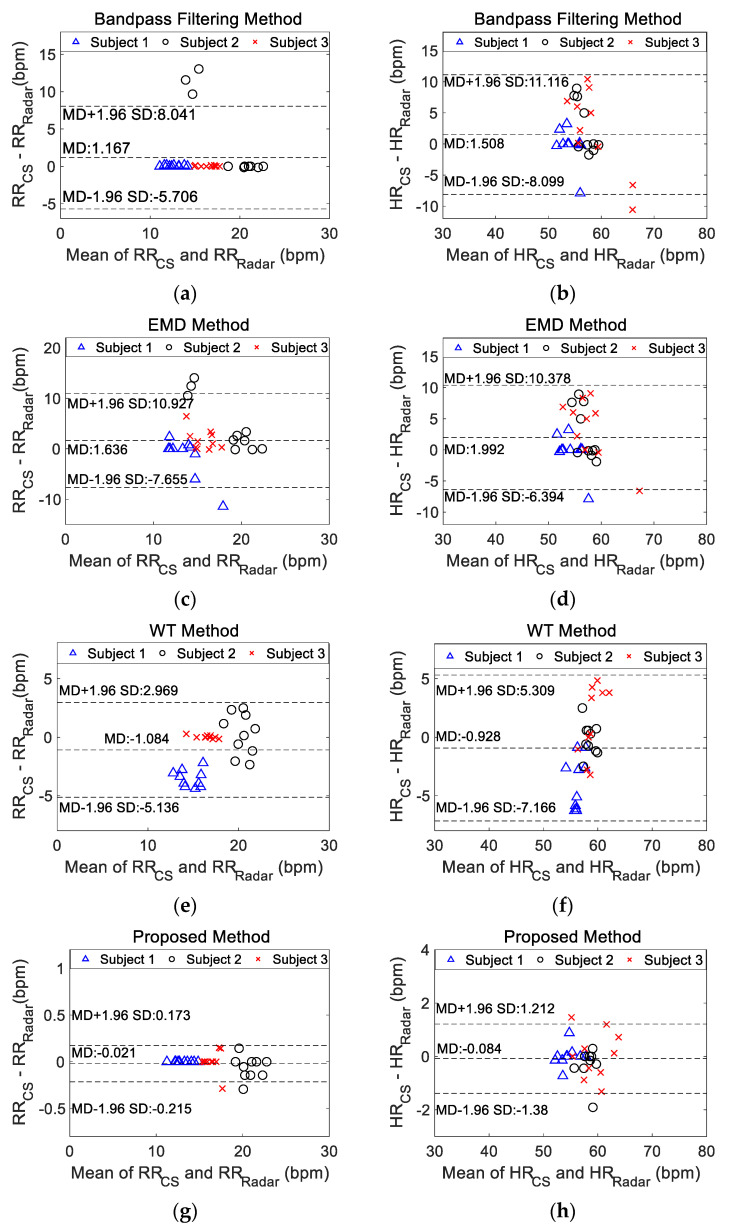
Bland–Altman plots showing the performance of the physiological signals recorded from contact sensors and the vital signals extracted from radar using different methods for the three volunteers, including 10 measurements. Bland–Altman plots for (**a**) the respiratory rate based on the bandpass filtering method, (**b**) the heartbeat rate based on the bandpass filtering method, (**c**) the respiratory rate based on the EMD method, (**d**) the heartbeat rate based on the EMD method, (**e**) the respiratory rate based on the WT method, (**f**) the heartbeat rate based on the WT method, (**g**) the respiratory rate based on the proposed method and (**h**) the heartbeat rate based on the proposed method. RR_CS_: the respiratory rate measured by contact sensors; RR_Radar_: the respiratory rate measured by radar; HR_CS_: the heartbeat rate measured by contact sensors; HR_Radar_: the heartbeat rate measured by radar; MD: the mean difference between the value measured from contact sensors and the value calculated by different methods from radar; SD: standard deviation of the differences between the measured value and the calculated value by different methods; EMD: empirical mode decomposition method; WT: wavelet transform method.

**Table 1 sensors-20-04913-t001:** Error estimation of respiratory rate (RR) and heartbeat rate (HR) of three volunteers for different methods (unit of all values is bpm).

Vital Signs	Statistical Parameters	Method			
		BPF	EMD	WT	Proposed
RR	MD	−1.167	−1.636	1.084	0.021
	SD	3.507	4.740	2.068	0.099
	MD + 1.96SD	5.706	7.655	5.136	0.215
	MD − 1.96SD	−8.041	−10.927	−2.969	−0.173
HR	MD	−1.508	−1.992	0.928	0.084
	SD	3.182	4.279	3.182	0.661
	MD + 1.96SD	4.729	6.394	7.166	1.380
	MD − 1.96SD	−7.746	−10.378	−5.309	−1.212

MD: the mean difference between the value measured from contact sensors and the value calculated by different methods from radar; SD: standard deviation of the differences between the measured value and the calculated value by different methods; RR: respiratory rate; HR: heartbeat rate; BPF: bandpass filtering method; EMD: empirical mode decomposition method; WT: wavelet transform method.

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
