# Peer review of "Adaptive Separation of Respiratory and Heartbeat Signals among Multiple People Based on Empirical Wavelet Transform Using UWB Radar"

_sensors, 2020, doi:10.3390/s20174913_

Round 1

Reviewer 1 Report

REVIEWER

This paper is related to vital signs monitoring using UWB radar. The main contribution of this paper is that it automatically separates heart and respiration signals using Empirical wavelets transform for multiple humans. The performance of radar based vital signs detection was compared with contact sensors. Following are the comments:

  1. On line 237-239, it is written that the range between radar and each individual is different for distinguishing between the individuals. Can you please give the minimum separation distance for detecting the vital signs of multihuman accurately? Although a theoretical range resolution of the radar is given in the paper but I am more interested in real experimental value as different subjects have different radar cross section values and vital signal is very sensitive to movements in the surrounding.
  2. What is meant by boundaries of EWT? Does it mean the frequency bands for respiration and heart beat signal? Are these boundaries pre-defined or it is automatically extracted from the radar data in real time. Please explain it in detail.
  3. Table 1 and Fig. 10 shows the superiority of the proposed method as compared to BPF, WT and EMD methods for three individuals. I’m curious that is the proposed method specifically outperforming the other algorithms for multiple humans or it is also better in terms of accuracy for single individual as well.
  4. The author main work is regarding separation of breathing and heart rate. However, an important reference with almost similar work (in terms of problem; not the solution) is missing in the literature section of introduction of this paper. It is suggested show how this work is better compared to the following work related to heart and breathing signal separation based on statistical properties of the vital signal.

Author Response

Dear Reviewer,

Thank you for your suggestions!

We have revised the paper according to your suggestions. Please download the word response for more details.

Best Regards,

Mi He

Reviewer 2 Report

This manuscript focus on a study for automated separation of respiratory and heartbeat signals based on empirical wavelet transform (EWT) for multiple people. It is well written. I recommend publishing it after a minor revision. The following are my suggestions and comments about this manuscript:

  1. You mentioned that EEMD does not always provide a decomposition of signals into components of desired frequency bands. Do you have test the CEEMD or VMD method in your dataset?
  2. Please add the description of fig.4a and 4b in the label.
  3. In your experiment, there is not between the radar system and human. How about the reliability of the UWB system in through-wall detection?
  4. It would help if you revised the fig.7. The number is not in-line. Besides, please use the black color for all curves in your manuscript.
  5. There is a distance range of three subjects in figure.4. The EWT components in figure 7, how can you extract the signal, sum the target range together, or select the maximum amplitude signal?
  6. In figure 8, the y-label is the amplitude (cm). What is the meaning of cm?

Author Response

(The authors gave the same response as above.)

Round 2

Reviewer 1 Report

I would like to recommend this paper for publication in the journal "sensors" in present form.